# PARP1 inhibitors trigger innate immunity via PARP1 trapping-induced DNA damage response

Chiho Kim, Xu-Dong Wang, Yonghao Yu*

Department of Biochemistry, University of Texas Southwestern Medical Center, Dallas, United States

**Abstract** It is being increasingly appreciated that the immunomodulatory functions of PARP1 inhibitors (PARPi) underlie their clinical activities in various *BRCA*-mutated tumors. PARPi possess both PARP1 inhibition and PARP1 trapping activities. The relative contribution of these two mechanisms toward PARPi-induced innate immune signaling, however, is poorly understood. We find that the presence of the PARP1 protein with uncompromised DNA-binding activities is required for PARPi-induced innate immune response. The activation of cGAS-STING signaling induced by various PARPi closely depends on their PARP1 trapping activities. Finally, we show that a small molecule PARP1 degrader blocks the enzymatic activity of PARP1 without eliciting PARP1 trapping or cGAS-STING activation. Our findings thus identify PARP1 trapping as a major contributor of the immunomodulatory functions of PARPi. Although PARPi-induced innate immunity is highly desirable in human malignancies, the ability of 'non-trapping' PARP1 degraders to avoid the activation of innate immune response could be useful in non-oncological diseases.

*For correspondence:
yonghao.yu@utsouthwestern.edu

## Introduction

Poly-ADP-ribose polymerase 1 (hereafter referred to as PARP1) is an enzyme that is critically involved in mediating DNA damage response (DDR). Upon sensing the genotoxic stress, PARP1 is recruited to DNA stand breaks and is activated to synthesize negatively charged Poly-ADP-ribose (PAR) polymers. One of the functions of these PAR chains is to serve as a platform to recruit the DDR machinery to repair and resolve these DNA breaks (*Ray Chaudhuri and Nussenzweig, 2017*; *Gibson and Kraus, 2012*). Therapeutics that target PARP1 have been proposed as an attractive strategy to treat human malignancies. Indeed, cancers with *BRCA1/2* mutations rely on PARP1 for genome integrity, and they are selectively killed by PARP1 inhibitors (PARPi) via the 'synthetic lethality' mechanism (*Lord et al., 2015*; *Lord and Ashworth, 2017*; *Farmer et al., 2005*). Four PARPi (Olaparib, Rucaparib, Niraparib, and Talazoparib) have been approved by the FDA to treat *BRCA1/2*-deficient breast and/or ovarian cancers (*Faraoni and Graziani, 2018*). In addition, PARPi are being extensively evaluated in the clinic, either as single agents or in combination with chemo- and radiation-therapy approaches, for the treatment of many other solid tumors (*Rouleau et al., 2010*; *Lord and Ashworth, 2017*).

All FDA-approved PARPi are $NAD^+$-competitive, and it was initially thought that these agents kill tumors simply by inhibiting the catalytic activity of PARP1. However, recent studies suggest that the cytotoxicity of PARPi is ascribed, at least in part, to the ability of these compounds to induce PARP1 trapping (*Hopkins et al., 2019*; *Murai et al., 2012*). During DDR, PARP1 is activated to catalyze the Poly-ADP-ribosylation (PARylation) of many proteins, including PARP1 itself. PARylation triggers the release of PARP1 from the DNA lesions, owing to the charge repulsion and steric hindrance introduced by the PAR polymers. PARPi block the synthesis of PAR chains, which causes PARP1 to be trapped on the chromatin. The trapped PARP1 triggers further DNA damage, cell cycle arrest, and

eventually, cancer cell death (*Lord and Ashworth, 2017*; *Slade, 2020*). Besides the PARylation-dependent mechanism, several recent studies also suggest that although the various clinically relevant PARPi all bind to PARP1, they induce different degrees of PARP1 conformational changes, and in doing so, PARP1 trapping (*Lord and Ashworth, 2017*; *Hopkins et al., 2019*; *Murai et al., 2014*; *Shen et al., 2013*; *Murai et al., 2012*).

Many recent studies have provided compelling evidence for a functional link between tumor DNA damage and the immune system, during the treatment of cancers. During chemo- and radiation-therapy, self-DNA is released, and is detected by the cytosolic DNA sensor, cyclic GMP-AMP (cGAMP) synthetase (cGAS). cGAS subsequently produces the second messenger cGAMP. cGAMP binds to Stimulator of Interferon Genes (STING), leading to the recruitment and activation of Tank-binding kinase I (TBK1). TBK1 phosphorylates a transcription factor called interferon regulatory factor 3 (IRF3), resulting in its nuclear translocation, and the IRF3-dependent activation of type I interferon (IFN) signaling (*Chen et al., 2016*; *Ishikawa and Barber, 2008*; *Li and Chen, 2018*; *Barber, 2015*). Thus, the cGAS-STING pathway plays a vital role not only in protecting the cells against a variety of pathogens, but also in the antitumor immune response. Because PARPi treatment is known to produce cytosolic dsDNA (double-stranded DNA), it has been proposed that the activation of innate immune signaling could be a critical molecular mechanism underlying the therapeutic effect of PARPi (*Ding et al., 2018*; *Shen et al., 2019*; *Pantelidou et al., 2019*; *Sen et al., 2019*). However, the relative contribution of the two independent, yet interconnected mechanisms (i.e., PARP1 inhibition and PARP1 trapping) in mediating the antitumor immunity of PARPi is not well understood.

In this study, we show that PARPi treatment induces the antitumor immune response via the cGAS-STING pathway. However, PARPi treatment generates cytosolic dsDNA, only in the presence of the PARP1 protein. PARPi-induced dsDNA is subsequently recognized by cGAS, which leads to the activation of innate immune signaling. We subsequently employed a series of clinically relevant PARPi with different PARP1 trapping activities, as well as a 'non-trapping' PARP1 degrader. We showed that the activation of innate immune signaling is critically dependent on the PARP1 trapping activity of these compounds. These results provide evidence that PARPi-mediated PARP1 trapping, but not the catalytic inhibition of PARP1, is a key determinant for the activation of the innate immune response.

## Results

### PARPi activates innate immune signaling via the cGAS-STING pathway

It is being increasingly appreciated that chemo- and radiation-therapy cause the formation of cytosolic dsDNA and micronuclei, which, in turn, lead to the activation of the cGAS-STING signaling pathway and inflammatory responses in tumors (*Vanpouille-Box et al., 2018*; *Liang and Peng, 2016*; *Harding et al., 2017*; *Mackenzie et al., 2017*; *Dou et al., 2017*; *Glück et al., 2017*; *Vanpouille-Box et al., 2017*; *Yum et al., 2019*). We explored the immunomodulatory functions of PARPi using Talazoparib, which is an FDA-approved PARP1 inhibitor that is known to potently inhibit and trap PARP1 (*Figure 1A*). We found that Talazoparib treatment was able to induce the formation of cytosolic dsDNA (*Figure 1B*) as well as γH2AX (a marker for DNA double strand breaks) (*Figure 1C*). To evaluate the innate immune response, we examined the phosphorylation of TBK1 (pS172 TBK1) and IRF3 (pS396 IRF3), two critical components in the cGAS-STING pathway (*Motwani et al., 2019*; *Kato et al., 2017*). Indeed, Talazoparib treatment dramatically increased both phosphorylation events (*Figure 1D and E*). Talazoparib treatment also remarkably induced the nuclear translocation of phospho-IRF3 (*Figure 1E*, right), which is a critical step for IRF3-mediated gene transcription (*Kato et al., 2017*; *Motwani et al., 2019*). We then examined the mRNA expression level of a number of known downstream target genes of the cGAS-STING pathway. Consistent with the previous studies (*Sun et al., 2013*; *Parkes et al., 2017*; *Fu et al., 2020*), Talazoparib treatment greatly upregulated the expression of type I interferons (IFN; *Inf-α* and *Inf-β*), pro-inflammatory cytokines (*Ccl5* and *Cxcl10*), and interferon-stimulated genes (ISGs; *Isg15*, *Mx1*, *Mx2*, and *Ifit3*) (*Figure 1F*, *Figure 1—figure supplement 1A*). To examine whether the cGAS-STING pathway is necessary for the PARPi-induced innate immune signaling, we depleted cGAS using two independent short hairpin RNAs (shRNAs) (*Figure 1G*). Knock-down (KD) of cGAS did not interfere with PARP1 trapping

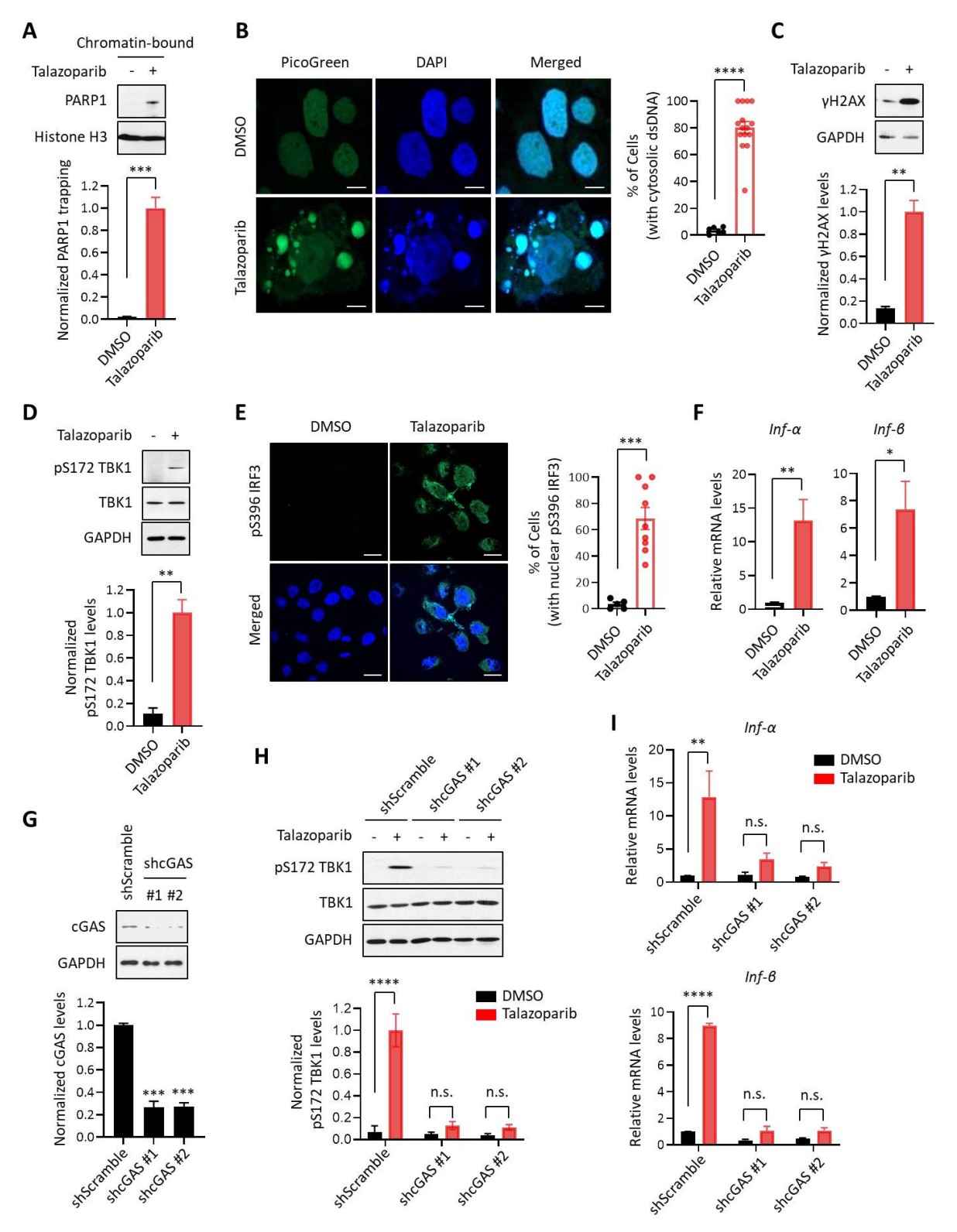

**Figure 1.** PARPi induces the innate immune response via the cGAS-STING pathway. (**A**) The level of trapped PARP1 in HeLa cells treated with or without Talazoparib (10 μM for 72 hr). Top, chromatin-bound fractions were isolated and were probed using the indicated antibodies. Histone H3 was used as the loading control. Bottom, the graph shows the quantification of the level of PARP1 trapping. Values were presented as means ± SD from three biological replicates. Significance was determined with unpaired Student's t-test. ***p < 0.001. (**B**) Staining of cytosolic dsDNA in HeLa cells

*Figure 1 continued on next page*

*Figure 1 continued*

treated with or without Talazoparib (10 μM for 72 hr). Left, representative image of PicoGreen (green) staining. DAPI (blue) was used to visualize the nucleus. Scale bars represent 10 μm. Right, the graph shows the quantification of the number of cells with cytosolic dsDNA. Values were presented as means ± SEM from three biological replicates (n = 3 fields,≥100 cells counted per condition). Significance was determined with unpaired Student's t-test. ****p < 0.0001. (C) The extent of DNA damage in HeLa cells treated with or without Talazoparib (10 μM for 72 hr). Top, whole cell lysates were probed using the indicated antibodies. Bottom, the graph shows the quantification of γH2AX levels. Values were presented as means ± SD from three biological replicates. Significance was determined with unpaired Student's t-test. **p < 0.01. (D) The level of pS172 TBK1 in HeLa cells treated with or without Talazoparib (10 μM for 72 hr). Top, whole cell lysates were probed using the indicated antibodies. Bottom, the graph shows the quantification of pS172 TBK1 levels. Values were presented as means ± SD from three biological replicates. Significance was determined with unpaired Student's t-test. **p < 0.01. (E) The level of pS396 IRF3 in HeLa cells treated with or without Talazoparib (10 μM for 72 hr). Left, representative image of pS396 IRF3 levels (green). DAPI (blue) was used to visualize the nucleus. Scale bars represent 20 μm. Right, the graph shows the quantification of the number of cells stained positive for pS396 IRF3 in nucleus. Values were presented as means ± SEM from three biological replicates (n = 3 fields,≥100 cells counted per condition). Significance was determined with unpaired Student's t-test. ***p < 0.001. (F) RT-qPCR of type I interferons levels in HeLa cells treated with or without Talazoparib (10 μM for 72 hr). Values of *Inf-α* and *Inf-β* were presented as means ± SEM from three biological replicates. Significance was determined with unpaired Student's t-test. *p < 0.05, **p < 0.01. (G) Knock-down of cGAS. HeLa cells expressing the control shRNA (shScramble) or shcGAS (shcGAS #1 or #2) were probed using the indicated antibodies. Right, the graph shows the ratio of cGAS depletion. Values were presented as means ± SD from three biological replicates. Significance was determined with one-way ANOVA. ***p < 0.001. (H) Depletion of cGAS abolishes PARPi-induced activation of innate immune signaling. HeLa cells expressing shRNA against control (shScramble) or cGAS (shcGAS #1 or #2) were treated with or without Talazoparib (10 μM for 72 hr). The cells were lysed and were immunoblotted using the indicated antibodies. Values were presented as means ± SD from three biological replicates. Significance was determined with two-way ANOVA. ****p < 0.0001, n.s., not significant. (I) RT-qPCR analyses of type I interferons. HeLa cells expressing shRNA against control (shScramble) or cGAS (shcGAS #1 or #2) were treated with or without Talazoparib (10 μM for 72 hr). Values of *Inf-α* and *Inf-β* mRNA levels were presented as means ± SEM from three biological replicates. Significance was determined with unpaired Student's t-test. ****p < 0.0001, n.s., not significant.

The online version of this article includes the following figure supplement(s) for figure 1:

**Figure supplement 1.** Proteomic profiling of PARPi-induced innate immune response (Part 1).
**Figure supplement 2.** Proteomic profiling of PARPi-induced innate immune response (Part 2).

(*Figure 1—figure supplement 1B*) or the subsequent DDR (*Figure 1—figure supplement 1C*). However, the activation of the innate immune response, as assessed by the level of pS172 TBK1 and the cGAS-STING target genes, was dramatically reduced in cGAS-depleted cells (*Figure 1H and I*, *Figure 1—figure supplement 1D*). Taken together, these results demonstrate that PARPi treatment induces the innate immune response via the cGAS-STING pathway.

To examine the immunomodulatory effects of PARPi in an unbiased manner, we performed isobaric labeling-based, global protein expression analysis in Talazoparib-treated MHH-ES-1 cells (an Ewing's sarcoma cell line that is highly sensitive to PARPi) (*Gill et al., 2015*). Talazoparib treatment was able to induce potent PARP1 trapping, γH2AX formation and TBK1 phosphorylation in this cell line (*Figure 1—figure supplement 2E–G*). Cells treated with DMSO or Talazoparib were lysed, and the proteins were digested with the resulting peptides labeled with the corresponding tandem mass tag (TMT) reagents. From this dataset, we were able to identify and quantify a total of 9545 proteins (protein false-discovery rate (FDR) < 1%) (*Supplementary file 1*). Correlation analysis revealed an excellent reproducibility between the biological replicate samples (*Figure 1—figure supplement 2H*). Compared to control, a total of 270 and 395 proteins were up- and down-regulated by at least two-fold, respectively, in Talazoparib-treated cells (*Figure 1—figure supplement 2I*). Intriguingly, gene ontology (GO) analyses of the up-regulated proteins showed that these proteins were highly enriched with biological processes connected to innate immune signaling (e.g., type I interferon signaling pathway, p=2.79 × 10⁻⁵ and immune response p=8.04 × 10⁻⁵), which we validated using independent RT-qPCR assays (*Figure 1—figure supplement 2J and K*, *Supplementary file 2*). For example, we identified Interleukin 1α (IL1A) as one of the Talazoparib-induced, up-regulated cytokines in our quantitative proteomic dataset. Furthermore, Talazoparib treatment also induced the coordinated upregulation of several immune signaling proteins, including ISG15, IFIT3, MX1 and MX2 (*Supplementary file 1*).

## PARP1 trapping is required for the PARPi-induced innate immune signaling

Because all FDA-approved PARPi possess both PARP1 trapping and PARP inhibition activities, we used a genetic method to assess their relative contribution to PARPi-induced activation of the cGAS-

STING pathway. Specifically, we generated PARP1 knock-out (KO) HeLa cells (*Figure 2—figure supplement 1A*) and found that Talazoparib treatment only induced PARP1 trapping in the wild-type (WT) cells, but not in PARP1 KO cells (*Figure 2A*). Accordingly, DDR, as detected by γH2AX, was also only elevated in the PARP1 WT cells (*Figure 2B*). Next, we evaluated whether the deletion of the PARP1 protein affects the PARPi-induced activation of the cGAS-STING pathway. As we expected, Talazoparib treatment led to a dramatic increase of pS172 TBK1 only in the PARP1 WT cells, but not PARP1 KO cells (*Figure 2C*). Talazoparib-induced IRF3 phosphorylation and its nuclear translocation were also blocked by PARP1 deletion (*Figure 2D*). Finally, PARP1 deletion also greatly diminished Talazoparib-induced expression of cGAS-STING target genes (*Figure 2E*, *Figure 2—figure supplement 1B*). These data indicate that the PARP1 protein is required for the PARPi-mediated activation of innate immune signaling.

We surveyed a series of clinically relevant PARPi, including Talazoparib, Niraparib, Rucaparib, Olaparib, and Veliparib. Consistent with previous studies, these compounds all potently blocked the enzymatic activity of PARP1 (*Figure 3—figure supplement 1A*). However, these compounds were able to induce different levels of PARP1 trapping (PARP1 trapping levels: Talazoparib > Niraparib > Rucaparib ≈ Olaparib > Veliparib) (*Figure 3A*). Interestingly, DDR as measured by γH2AX was correlative with respect to the level of PARP1 trapping elicited by these PARPi (DNA damage levels: Talazoparib > Niraparib > Rucaparib ≈ Olaparib > Veliparib) (*Figure 3B*). Accordingly, the cytotoxicity of these compounds also positively correlated with their PARP1 trapping activities (*Figure 3—figure supplement 1D*). Finally, the activation of the cGAS-STING pathway, as measured by the pS172 TBK level, also correlated with PARP1 trapping (*Figure 3C*). As an example, compared to Rucaparib, Talazoparib was able to induce a much stronger activation of the cGAS-STING pathway (*Figure 3—figure supplement 1B*). Taken together, these results showed that the level of PARP1 trapping, DNA damage, cytotoxicity and cGAS-STING activation was all positively correlated for the various PARPi (*Figure 3D*).

To further explore the role of PARP1 trapping in mediating the innate immune response of the PARPi, we employed a PARP1 mutant (R138C) that was identified from a chemical-induced mutagenesis screen performed in mouse embryonic stem cells (mESCs) (*Herzog et al., 2018*). This PARP1 mutant bears a significantly reduced DNA binding capability, and as a result, it cannot be trapped on the chromatin upon the treatment of PARPi. We generated PARP1 KO cells, and reconstituted these cells using either WT PARP1 or the PARP1 R138C mutant. Talazoparib treatment dramatically elevated the levels of PARP1 trapping in WT PARP1-reconsistuted cells, but not in cells reconstituted with the PARP1 R138C mutant (*Figure 3E*). Cells expressing the PARP1 R138C mutant also had greatly reduced DDR, upon Talazoparib treatment (*Figure 3F*). Finally, the expression of the PARP1 R138C mutant also prevented Talazoparib-induced activation of the cGAS-STING pathway (*Figure 3F and G*, *Figure 3—figure supplement 1C*). These results strongly supported the notion that PARP1 trapping is a prerequisite for the PARPi-induced activation of innate immune signaling.

## PARP1 degraders block PARP1 without eliciting PARP1 trapping or the subsequent innate immune signaling

Using the Proteolysis Targeting Chimera (PROTAC) strategy, we recently developed a series of small molecule compounds that selectively degrade PARP1 (*Wang et al., 2019*). These compounds were derived by linking a PARPi (e.g., Rucaparib) and an E3 binder (e.g., pomalidomide) by a covalent chemical linker. Unlike regular PARPi, these compounds block both the enzymatic and scaffolding effects of PARP1, and thereby could dissect PARP1 inhibition vs. PARP1 trapping. Consistent with the notion that PARP1 trapping is a major contributor of the PARPi-induced cytotoxicity (*Wang et al., 2019*), the treatment of HeLa cells using one such compound (iRucaparib-AP6) led to robust downregulation of PARP1 in HeLa cells. In contrast, the parent compound (Rucaparib) only induced the cleavage, but not the degradation of PARP1, presumably because of its toxicity in these cells (*Figure 4A*). Using a different cell line system (i.e., MHH-ES-1), we also showed that Rucaparib, but not iRucaparib-AP6, strongly promoted cell death (*Figure 4—figure supplement 1*).

Consistent with the diminished pool of total PARP1, iRucaparib-AP6 treatment resulted in minimal PARP1 trapping and γH2AX formation (*Figure 4B and C*). Accordingly, the level of pS172 TBK1 were dramatically increased in Rucaparib-treated, but not in iRucaparib-AP6-treated cells (*Figure 4D*). We examined the expression of cGAS-STING target genes in these cells, and found

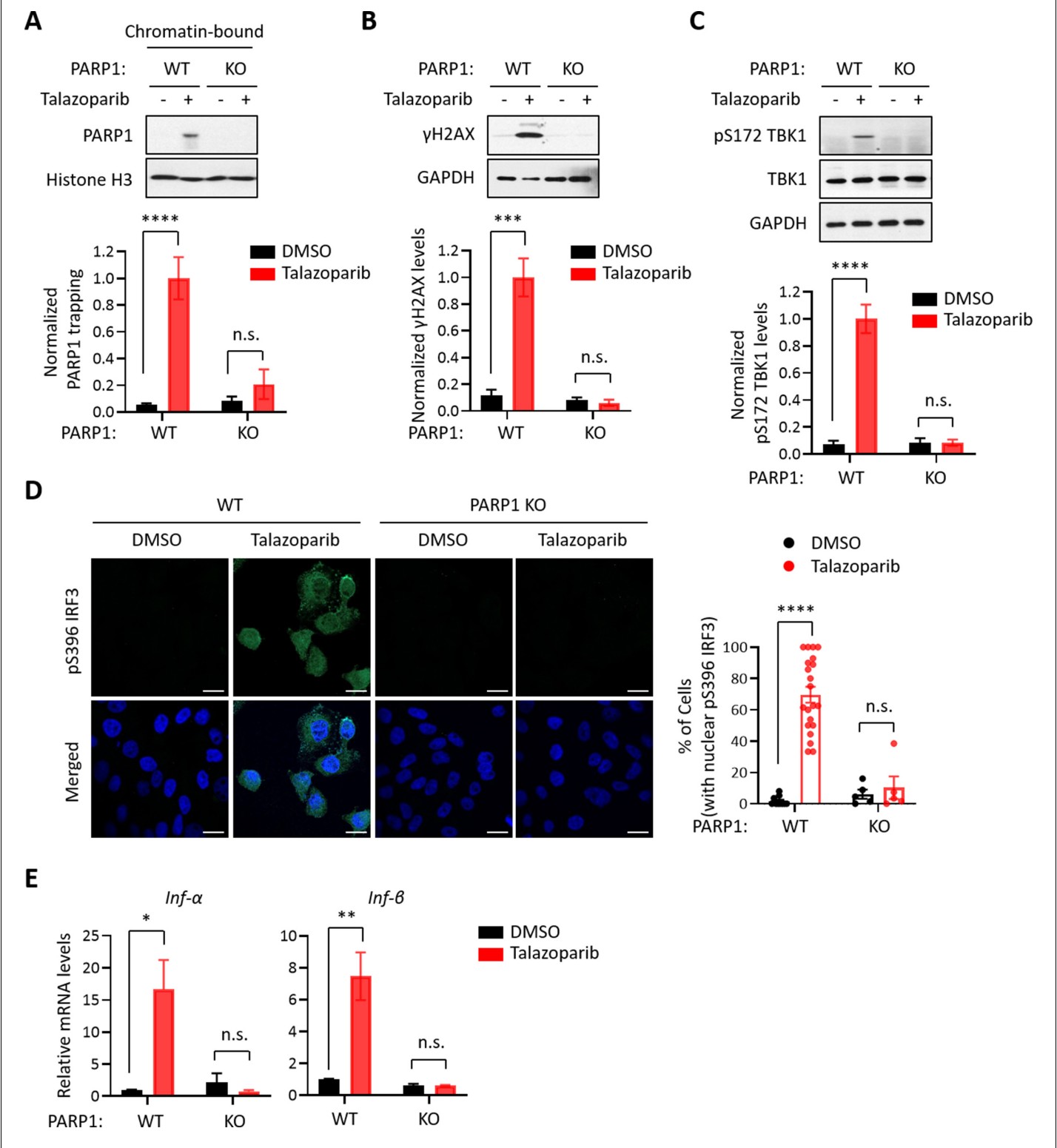

**Figure 2.** The PARP1 protein is required for PARPi-induced innate immune signaling. (**A**) PARPi-induced PARP1 trapping in wild-type (WT) and PARP1 knockout (KO) HeLa cells. Cell were also treated with or without Talazoparib (10 μM for 72 hr). Top, chromatin-bound fractions were isolated and were probed using the indicated antibodies. Histone H3 was used as the loading control. Bottom, the graph shows the quantification of the level of PARP1 trapping. Values were presented as means ± SD from three biological replicates. Significance was determined with two-way ANOVA. ****p < 0.001, n.s., not significant. (**B**) DDR in WT and PARP1 KO HeLa cells treated with or without Talazoparib (10 μM for 72 hr). Top, whole cell lysates were probed

*Figure 2 continued on next page*

*Figure 2 continued*

using the indicated antibodies. Bottom, the graph shows the quantification of γH2AX levels. Values were presented as means ± SD from three biological replicates. Significance was determined with two-way ANOVA. $^{***}p < 0.001$, n.s., not significant. (C) The level of pS172 TBK1 in WT and PARP1 KO HeLa cells treated with or without Talazoparib (10 μM for 72 hr). Top, whole cell lysates were probed using the indicated antibodies. Bottom, the graph shows the quantification of pS172 TBK1 levels. Values were presented as means ± SD from three biological replicates. Significance was determined with two-way ANOVA. $^{****}p < 0.0001$, n.s., not significant. (D) Staining of pS396 IRF3 levels in WT and PARP1 KO HeLa cells treated with or without Talazoparib (10 μM for 72 hr). Left, a representative image of pS396 IRF3 levels (green). DAPI (blue) was used to visualize the nucleus. Right, the graph shows the quantification of the number of cells stained positive for pS396 IRF3 in the nucleus. Values were presented as means ± SEM from three biological replicates. Significance was determined with two-way ANOVA. $^{****}p < 0.0001$, n.s., not significant. (E) RT-qPCR analyses of type I interferons in WT and PARP1 KO HeLa cells treated with or without Talazoparib (10 μM for 72 hr). Values of *Inf-α* and *Inf-β* mRNA levels were presented as means ± SEM from three biological replicates. Significance was determined with two-way ANOVA. $^{*}p < 0.05$, $^{**}p < 0.01$, n.s., not significant.

The online version of this article includes the following figure supplement(s) for figure 2:

**Figure supplement 1.** The PARP1 protein is required for PARPi-induced innate immune response.

that Rucaparib, but not iRucaparib-AP6, treatment significantly elevated the mRNA levels of type I IFNs, pro-inflammatory cytokines and ISGs (*Figure 4E*).

## Discussion

Since Rudolf Virchow observed the possible link between the immune system and tumors using lymphoid cells in a tumor in 1863, to use the immune system promoting antitumor response has been confirmed as one of the major breakthroughs in oncology, yielding the possibility of long-term clinical benefit and prolonged survival (*Zitvogel et al., 2008*; *Swann and Smyth, 2007*). The innate immune system as one of antitumor immune responses is composed of molecules and cells that respond to external and internal danger signals such as pathogen-associated molecular patterns (PAMPs) and damage-associated molecular patterns (DAMPs). PAMPs and DAMPs bind to their respective pattern recognition receptors (PRRs) to initiate immune responses. Thus, cytosolic PRRs including nucleotide-binding oligomerization domain–like receptors, retinoic acid–inducible gene I–like receptors (RLRs), and cGAS detect intracellular pathogens (*Wu and Chen, 2014*). Upon ligand binding, PRRs activate downstream signaling cascades to induce inflammatory responses such as the innate immune response, providing early protection against pathogen invasion or cellular damage.

Unlike normal cells, cancer cells are often replete with cytosolic dsDNA that originates from genomic, mitochondrial, and exogenous sources (*Vanpouille-Box et al., 2018*). Accumulating data have been reported that acute genomic stressors, including radiation, cisplatin, and intrinsic DNA damage generate cytosolic dsDNA and micronuclei to activate cGAS–STING in cancer cells (*Ahn et al., 2014*; *Harding et al., 2017*; *Mackenzie et al., 2017*; *Dou et al., 2017*). The role of PARPi as an inducer of DNA damage response has been well established to explain the cytotoxic effects of these compounds. However, accumulating evidence have pointed out that coordinated activation of both local and systemic antitumor immune responses could also underlie the antitumor effects of PARPi (*Pantelidou et al., 2019*; *Ding et al., 2018*; *Chabanon et al., 2019*).

Consistent with these previous studies, our date showed that PARPi treatment results in the robust production of cytosolic dsDNA, which leads to the subsequent activation of cGAS-STING signaling and the downstream innate immune pathway. The current PARPi are known to kill tumors via two distinct, but interconnected, mechanisms (i.e., PARP1 inhibition vs. PARP1 trapping) (*Zhen and Yu, 2018*; *Zhang et al., 2013*; *Wang et al., 2019*). The relative contribution of these two mechanisms in PARPi-mediated innate immune signaling, however, is poorly understood. Here, we sought to address this important question by using several independent systems. First, we found that the PARP1 protein is required for the PARPi-induced activation of cGAS-STING signaling. Indeed, Talazoparib was unable to cause PARP1 trapping, DDR and TBK1 activation in PARP1 KO cells. It is also important to note that the deletion of PARP1 alone does not lead to TBK1 activation and the expression of inflammatory genes, suggesting that the blockage of PARP1 catalytic activity is not sufficient to drive the activation of cGAS-STING signaling. Second, we found that the DNA-binding activity of PARP1 is required for the PARPi-induced activation of cGAS-STING signaling. Specifically, we employed a recently described PARP1 mutant (R138C) that was identified from an EMS-induced

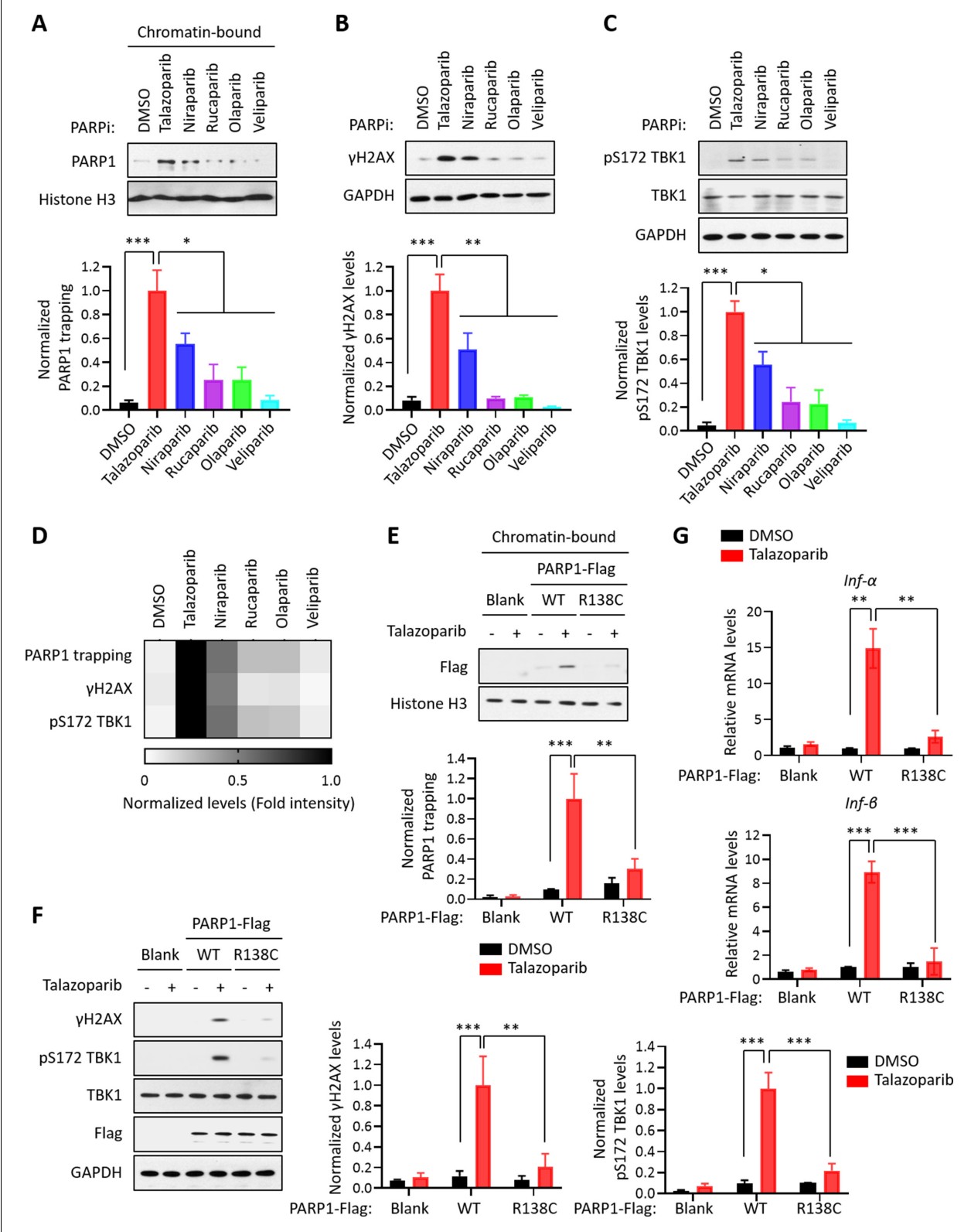

**Figure 3.** PARP1 trapping is the major contributor of PARPi-induced innate immune signaling. (**A**) The level of trapped PARP1 in HeLa cells treated with or without the indicated PARPi (10 µM for 72 hr). Top, chromatin-bound fractions were isolated and were probed using the indicated antibodies. Histone H3 was used as the loading control. Bottom, the graph shows the quantification of the level of PARP1 trapping. Values were presented as means ± SD from three biological replicates. Significance was determined with one-way ANOVA. *p < 0.05, ***p < 0.001. (**B**) The extent of DNA damage

*Figure 3 continued on next page*

*Figure 3 continued*

in HeLa cells treated with or without the indicated PARPi (10 μM for 72 hr). Top, whole cell lysates were probed using the indicated antibodies. Bottom, the graph shows the quantification of γH2AX levels. Values were presented as means ± SD from three biological replicates. Significance was determined with one-way ANOVA. **p < 0.01, ***p < 0.001. (C) The level of pS172 TBK1 in HeLa cells treated with or without the indicated PARPi (10 μM for 72 hr). Top, whole cell lysates were probed using the indicated antibodies. Bottom, the graph shows the quantification of pS172 TBK1 levels. Values were presented as means ± SD from three biological replicates. Significance was determined with one-way ANOVA. *p < 0.05, ***p < 0.001. (D) Heatmap of PARP1 trapping, DNA damage, and pS172 TBK1 levels for each PARPi. The normalized levels of PARP1 trapping (A), γH2AX (B), and pS172 TBK1 (C) are shown. (E) PARPi does not induce the trapping of a PARP1 mutant with defective DNA binding. Top, HeLa PARP1 KO cells expressing WT PARP1 or R138C mutant PARP1 (R138C) were treated with or without Talazoparib (10 μM for 72 hr). Chromatin-bound fractions were isolated and were probed using the indicated antibodies. Histone H3 was used as the loading control. Bottom, the graph shows the quantification of the levels of PARP1 trapping. Values were presented as means ± SD from three biological replicates. Significance was determined with two-way ANOVA. **p < 0.01, ***p < 0.001. (F) The extent of DNA damage in HeLa PARP1 KO cells expressing WT PARP1 or R138C PARP1 that were treated with or without Talazoparib (10 μM for 72 hr). Left, whole cell lysates were probed using the indicated antibodies. Right, the graph shows the quantification of γH2AX and pS172 TBK1 levels. Values were presented as means ± SD from three biological replicates. Significance was determined with two-way ANOVA. **p < 0.01, ***p < 0.001. (G) RT-qPCR analyses of type I interferons in HeLa PARP1 KO cells expressing WT or R138C PARP1 that were treated with or without Talazoparib (10 μM for 72 hr). Values of *Inf-α* and *Inf-β* mRNA levels were presented as means ± SEM from three biological replicates. Significance was determined with two-way ANOVA. **p < 0.01, ***p < 0.001.

The online version of this article includes the following figure supplement(s) for figure 3:

**Figure supplement 1.** PARP1 trapping is required for PARPi-induced cytotoxicity and innate immune response.

random mutagenesis screen for resistance to a PARPi (i.e., Olaparib) (*Herzog et al., 2018*). This mutant is defective for DNA binding and PARP1 trapping, and mouse embryonic stem cells (mESCs) that bear this mutation are resistant to Olaparib. We generated PARP1-deleted cells and reconstituted them with either WT PARP1 or the PARP1 R138C mutant. We found that Talazoparib was able to induce DDR and cGAS-STING signaling only in cells expressing WT PARP1, but not the PARP1 R138C mutant. Third, we utilized a panel of 5 clinically relevant PARPi (i.e., Talazoaprib, Niraparib, Rucaparib, Olaparib, and Veliparib) (*Lord and Ashworth, 2017*). While these compounds all potently blocked the formation of PAR, they displayed a dramatically different capability in inducing PARP1 trapping. Indeed, a recent study showed that the different structural elements within these PARPi drive, in an allosteric manner, the release or retention of these compounds at a DNA break (*Zandarashvili et al., 2020*). Intriguingly, these compounds then induce DDR and activation of cGAS-STING signaling that is closely correlated with the degree of PARP1 trapping (*Figure 3A–D*, *Figure 3—figure supplement 1A and D*). Fourth, we utilized that a recently developed, 'non-trapping' PARP1 degrader (iRucaparib-AP6) (*Wang et al., 2019*). This compound is cell-membrane permeable and is able to block the enzymatic activity of PARP1. However, unlike regular PARPi, iRucaparib-AP6 degrades PARP1, which prevents PARP1 trapping, DDR, cytotoxicity, and finally, the activation of cGAS-STING signaling (*Figure 4*, *Figure 4—figure supplement 1*). Finally, recent studies have pointed to cytosolic dsDNA as a source of stimulus for the activation of cGAS-STING pathway (*Sun et al., 2013*; *Wu et al., 2014*; *Civril et al., 2013*). In the context of PAPRi, cytosolic dsDNA could result from PARPi-induced PARP1 trapping, and subsequently, stalled or collapsed replication forks. The detailed molecular underpinnings of this critical pathway need to be addressed in future studies.

In conclusion, we have identified a direct mechanism of the antitumor immune response of PARPi (*Figure 4F*). We demonstrated that the ability to induce PARP1 trapping is the primary driver for the PARPi-mediated cytotoxicity and activation of innate immune signaling in cancer cells. In the presence of PARP1, PARPi-induced PARP1 trapping generates cytosolic dsDNA, which activates cGAS, and the downstream innate immune response. Although the immunomodulatory roles of PARPi are highly desirable in human malignancies, the ability for PARP1 degraders to prevent PARP1 trapping, and hence the activation of innate immune response could be useful in other contexts (e.g., ischemia-reperfusion injury and neurodegenerative diseases). The full therapeutic potential of this class of compounds warrants further studies.

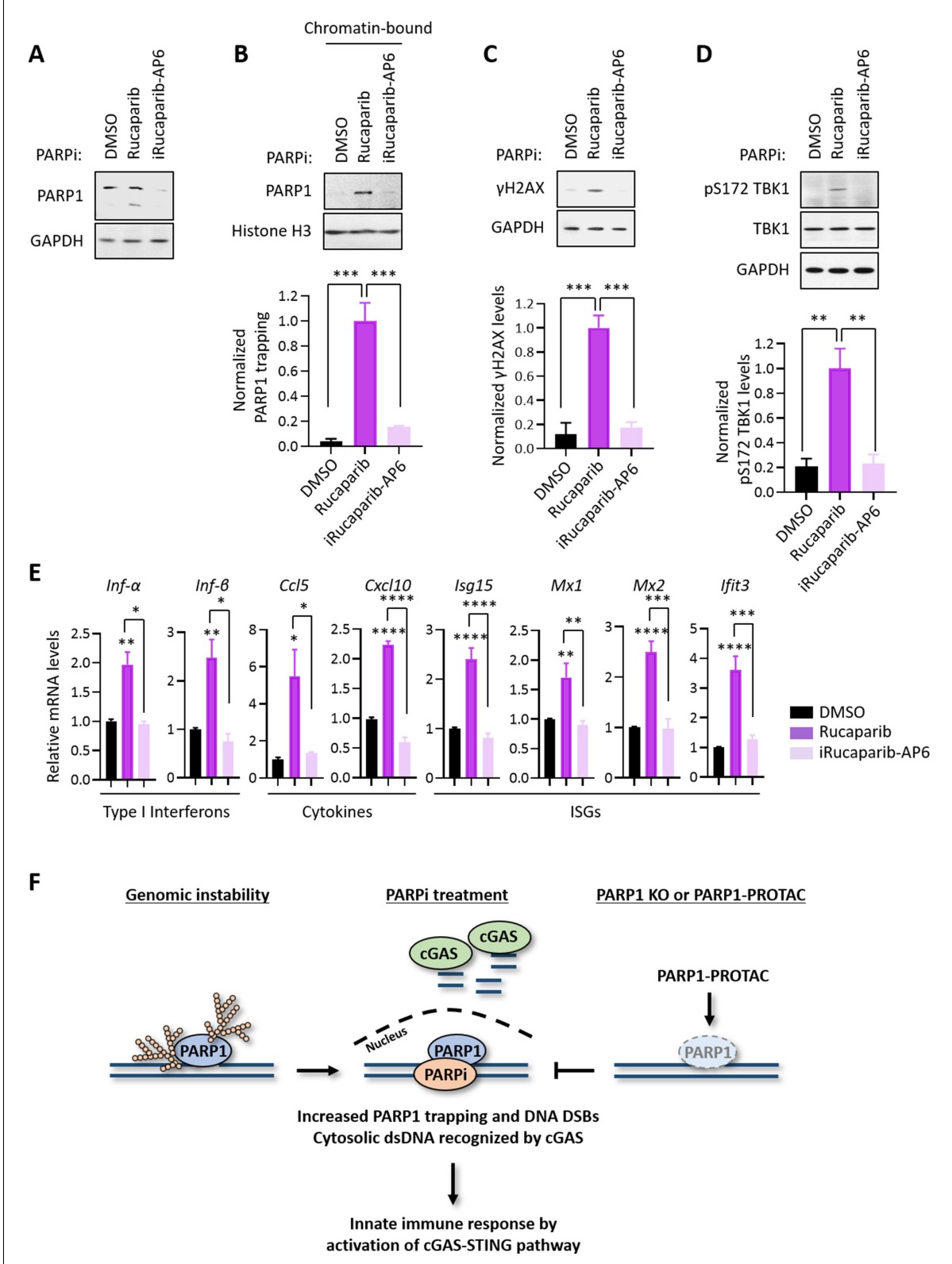

**Figure 4.** PARP1 degraders abolish PARP1-trapping induced innate immune signaling. (A) The level of PARP1 in HeLa cells treated with either Rucaparib or iRucaparib-AP6 (10 μM for 72 hr). Whole cell lysates were probed using the indicated antibodies. GAPDH was used as the loading control. (B) The level of trapped PARP1 in HeLa cells treated with either Rucaparib or iRucaparib-AP6 (10 μM for 72 hr). Top, chromatin-bound fractions were isolated and were probed using the indicated antibodies. Histone H3 was used as the loading control. Bottom, the graph shows the quantification of

*Figure 4 continued on next page*

*Figure 4 continued*

the level of PARP1 trapping. Values were presented as means ± SD from three biological replicates. Significance was determined with one-way ANOVA. [***]p < 0.001. (C) The extent of DNA damage in HeLa cells treated with either Rucaparib or iRucaparib-AP6 (10 µM for 72 hr). Top, whole cell lysates were probed using the indicated antibodies. Bottom, the graph shows the quantification of γH2AX levels. Values were presented as means ± SD from three biological replicates. Significance was determined with one-way ANOVA. [***]p < 0.001. (D) The level of pS172 TBK1 in HeLa cells treated with either Rucaparib or iRucaparib-AP6 (10 µM for 72 hr). Top, whole cell lysates were probed using the indicated antibodies. Bottom, the graph shows the quantification of pS172 TBK1 levels. Values were presented as means ± SD from three biological replicates. Significance was determined with one-way ANOVA. [**]p < 0.01. (E) RT-qPCR analyses of the cGAS-STING target gene expression in HeLa cells treated with either Rucaparib or iRucaparib-AP6 (10 µM for 72 hr). Values of type I interferons, cytokines, and ISGs mRNA levels were presented as means ± SEM from three biological replicates. Significance was determined with one-way ANOVA. [*]p < 0.05, [**]p < 0.01, [***]p < 0.001, [****]p < 0.0001. (F) The model of the activation of innate immune response via PARPi-induced PARP1 trapping.

The online version of this article includes the following figure supplement(s) for figure 4:

**Figure supplement 1.** PARP1 degraders prevent PARP1 trapping-induced cytotoxicity.

# Materials and methods

## Key resources table

| Reagent type (species) or resource | Designation | Source or reference | Identifiers | Additional information |
|---|---|---|---|---|
| Antibody | Anti-PAR (Mouse monoclonal) | Trevigen | Cat# 4335-MC-100, RRID:AB_2572318 | IB (1:3000) |
| Antibody | Anti-PARP1 (Rabbit polyclonal) | Cell Signaling Technology | Cat# 9542, RRID:AB_2160739 | IB (1:1000) |
| Antibody | Anti-γH2AX (Rabbit monoclonal) | Cell Signaling Technology | Cat# 9718, RRID:AB_2118009 | IB (1:3000) |
| Antibody | Anti-Histone H3 (Rabbit monoclonal) | Cell Signaling Technology | Cat# 4499, RRID:AB_10544537 | IB (1:3000) |
| Antibody | Anti-GAPDH (Mouse monoclonal) | Santa Cruz Biotechnology | Cat# sc-32233, RRID:AB_627679 | IB (1:5000) |
| Antibody | Anti-Phospho-TBK1/NAK (pS172 TBK1) (Rabbit monoclonal) | Cell Signaling Technology | Cat# 5483, RRID:AB_10693472 | IB (1:1000) |
| Antibody | Anti-TBK1/NAK (Rabbit monoclonal) | Cell Signaling Technology | Cat# 3504, RRID:AB_2255663 | IB (1:1000) |
| Antibody | Anti-Flag (Rabbit polyclonal) | MilliporeSigma | Cat# F7425, RRID:AB_439687 | IB (1:1000) |
| Antibody | Anti-cGAS (Rabbit monoclonal) | Cell Signaling Technology | Cat# 15102, RRID:AB_2732795 | IB (1:1000) |
| Antibody | Anti-Phospho-IRF-3 (pS396 IRF3) (Rabbit monoclonal) | Cell Signaling Technology | Cat# 4947, RRID:AB_823547 | IF (1:1000) |
| Antibody | Alexa Fluor 488-conjugated goat anti-rabbit IgG (Goat polyclonal) | Thermo Fisher Scientific | Cat# A32731, RRID:AB_2633280 | IF (1:1000) |
| Antibody | Goat Anti-Mouse IgG Antibody, HRP conjugate, Species Adsorbed (Goat polyclonal) | MilliporeSigma | Cat# AP181P, RRID:AB_11214094 | IB (1:3000) |
| Antibody | ECL Rabbit IgG, HRP-linked fragment (Donkey polyclonal) | GE Healthcare life sciences | Cat# NA9340, RRID:AB_772191 | IB (1:3000) |

*Continued on next page*

*Continued*

| Reagent type (species) or resource | Designation | Source or reference | Identifiers | Additional information |
|---|---|---|---|---|
| Chemical compound, drug | Talazoparib | Selleck | Cat# S7048 | PARP1 inhibitor |
| Chemical compound, drug | Niraparib | Selleck | Cat# S2741 | PARP1 inhibitor |
| Chemical compound, drug | Rucaparib | Selleck | Cat# S1098 | PARP1 inhibitor |
| Chemical compound, drug | Olaparib | Selleck | Cat# S1060 | PARP1 inhibitor |
| Chemical compound, drug | Veliparib | Selleck | Cat# S1004 | PARP1 inhibitor |
| Chemical compound, drug | iRucaparib-AP6 | Our laboratory | N/A | PARP1 degrader |
| Chemical compound, drug | Polybrene (Hexadimethrine bromide) | MilliporeSigma | Cat# H9268; CAS 28728-55-4 | |
| Chemical compound, drug | Puromycin | MilliporeSigma | Cat# P7255; CAS 58-58-2 | |
| Cell line (human) | HeLa | ATCC | Cat# CCL-2, RRID:CVCL_0030 | |
| Cell line (human) | HeLa PARP1 KO | In this study | N/A | PARP1 deficient HeLa |
| Cell line (human) | MHH-ES-1 | DSMZ | Cat# ACC 167, RRID:CVCL_1411 | |
| Cell line (human) | HeLa | ATCC | Cat# CCL-2 | |
| Recombinant DNA reagent | pCMV-hPARP1-3xFlag-WT | Addgene | Cat# 11157 | In pCMV; tagged with 3XFlag on its N-terminus |
| Recombinant DNA reagent | pCMV-hPARP1-3xFlag-R138C | In this study | Modified by R138C mutation | In pCMV; tagged with 3XFlag on its N-terminus |
| Strain, strain background (*Escherichia coli*) | DH5alpha | Thermo Fisher Scientific | Cat# 18258012 | Competent cells |
| Strain, strain background (*Escherichia coli*) | Stbl3 Competent *E. coli* | Thermo Fisher Scientific | Cat# C737303 | Competent cells |
| Commercial assay or kit | Quant-iT PicoGreen dsDNA Reagent | Thermo Fisher Scientific | Cat# P7581 | |
| Commercial assay or kit | Subcellular Protein Fractionation Kit for Cultured Cells | Thermo Fisher Scientific | Cat# 78840 | |
| Commercial assay or kit | RNeasy Mini Kit | QIAGEN | Cat# 74104 | |
| Commercial assay or kit | QIAprep Spin Miniprep Kit | QIAGEN | Cat# 27106 | |
| Commercial assay or kit | e-Myco PLUS Mycoplasma PCR Detection Kit | BOCA SCIENTIFIC | Cat# 25237 | |
| Commercial assay or kit | TMT6plex Mass Tag Labeling Kits | Thermo Fisher Scientific | Cat# 90110 | |
| Commercial assay or kit | CellTiter-Glo Luminescent Cell Viability Assay | Promega | Cat# G7571 | |
| Software, algorithm | ImageJ 1.49 v | NIH | https://imagej.net/ImageJ2 | |
| Software, algorithm | DAVID Bioinformatics Resources v6.8 | DAVID | https://david.ncifcrf.gov/ | |
| Software, algorithm | PRISM v8.2.0 | GraphPad | https://www.graphpad.com/scientific-software/prism/ | |

*Continued on next page*

*Continued*

| Reagent type (species) or resource | Designation | Source or reference | Identifiers | Additional information |
|---|---|---|---|---|
| Software, algorithm | human protein sequences (Uniprot) | UniProt | https://www.uniprot.org/UniProt database (2019_07,560,537 sequences;taxonomy, *Homo sapiens*, 20,431 ) | |
| Software, algorithm | human IPI protein database v3.60 | EMBL-EBI | ftp://ftp.ebi.ac.uk/pub/databases/IPI | |
| Software, algorithm | The Sequest algorithm v28 | Cell. 2010 Dec 3 ;143(7):1174–89 | N/A | |
| Others | DAPI | MilliporeSigma | Cat# D9542 | |
| Others | Dimethyl sulfoxide (DMSO) | Thermo Fisher Scientific | Cat# BP231-1; CAS 67-68-5 | |
| Others | Lipofectamine 2000 | Thermo Fisher Scientific | Cat# 11668500 | |
| Others | Dulbecco's Modified Eagle's Medium (DMEM) | MilliporeSigma | Cat# D5796 | |
| Others | RPMI1640 | MilliporeSigma | Cat# R8758 | |
| Others | Fetal Bovine Serum (FBS) | MilliporeSigma | Cat# 12303C | |
| Others | Lysyl Endopeptidase (Lys-C) | Wako | Cat# 129–02541; CAS 123175-82-6 | |
| Others | Gen5 | BioTek | N/A | |
| Others | BCA reagents | Thermo Fisher Scientific | Cat# 23224/23228 | |
| Others | 0.45 $\mu$m filter | Thermo Fisher Scientific | Cat# 05-713-387 | |
| Others | Synergy HT Multi-Detection Microplate Reader. | BioTek | N/A | |
| Others | SuperScript III Reverse Transcriptase | Thermo Fisher Scientific | Cat# 18080044 | |
| Others | CFX384 Touch Real-Time PCR Detection System | Bio-Rad | Cat# 1855484 | |
| Others | Applied Biosystems Power SYBR Green PCR Master Mix | Thermo Fisher Scientific | Cat# 43-676-59 | |
| Others | Oasis HLB solid-phase extraction (SPE) cartridges | Waters | Cat# 186000383 | |
| Others | 3M Empore C18 Extraction Disk | Thermo Fisher Scientific | Cat# 14-386-2 | |
| Others | ZORBAX 300Extend-C18 HPLC column | Agilent | Cat# 761775–902 | |
| Others | Q Exactive Hybrid Quadrupole-Orbitrap Mass Spectrometer | Thermo Fisher Scientific | Cat# IQLAAEGAAPFALGMAZR | |
| Others | PicoFrit nanospray columns | New Objective | PF360-75-15-N-5 | |

## Cell lines and culture procedures

Human cervical carcinoma cells (HeLa, from ATCC) were maintained in high glucose Dulbecco's Modified Eagle's Medium (DMEM) (MilliporeSigma), supplemented with 10% Fetal bovine serum (FBS) (MilliporeSigma) at 37℃ in 5% $CO_2$. Human Ewing's Sarcoma cells (MHH-ES-1, from DSMZ) were maintained in RPMI1640 (MilliporeSigma), supplemented with 10% Fetal bovine serum (FBS) (MilliporeSigma) at 37℃ in 5% $CO_2$. All cell lines have been DNA fingerprinted using the PowerPlex 1.2 kit (Promega) and are found to be mycoplasma free using the e-Myco kit (Boca Scientific). The concentrations and times of each chemical treatment are indicated in the figure legends.

## Antibodies and reagents

Antibodies against the following proteins were used. Cell Signaling Technology: PARP1 (#9542), $\gamma$H2AX (#9718), Histone H3 (#4499), Phospho-TBK1/NAK (pS172 TBK1; #5483), TBK1/NAK (#3504), cGAS (#15102), STING (#13647), Phospho-IRF-3 (pS396 IRF3; #4947); Santa Cruz Biotechnology: GAPDH (#sc-32233); Trevigen: PAR (#4335-MC-100); MilliporeSigma: Flag (#F7425). Thermo Fisher: Alexa Fluor 488-conjugated goat anti-rabbit IgG (Cat# A32731). The following reagents were used: Talazoparib, Niraparib, Rucaparib, Olaparib, and Veliparib were all purchased from Selleck; iRucaparib-AP6 was synthesized in previous our report (*Wang et al., 2019*). Dimethyl sulfoxide (DMSO), and Lipofectamine 2000 were all purchased from Thermo Fisher Scientific; Polybrene (Hexadimethrine bromide) and Puromycin were purchased from MilliporeSigma.

## Immunoblot analysis

Cellular lysates were prepared using a 1% SDS lysis buffer containing 10 mM HEPES, pH 7.0, 2 mM $MgCl_2$, 20 U/mL universal nuclease. Cellular lysates were clarified by centrifugation at 14,000 $\times$ g at 4°C for 15 min. Protein concentrations were determined with the BCA assay (Thermo Fisher Scientific). The resulting supernatants were subjected to immunoblot analysis with the corresponding antibodies. Enhanced chemiluminescence was used to detect specific bands using standard methods as previously described (*Kim et al., 2016*). The relative band intensity was measured using the Image J imaging software.

## Immunofluorescence

For immunofluorescence localization of the target molecules, HeLa PARP1 WT and KO cells were cultured on the cover glasses. Cells were fixed with 4% paraformaldehyde (Electron Microscopy Sciences, Hatfield, PA, USA) and blocked for 1 hr at RT in PBS (Lonza, Basel, Switzerland) containing 5% FBS and 0.2% Triton X-100. Cells were then incubated with a Rabbit monoclonal anti-pS396 IRF3 antibody overnight at 4°C, followed by incubation with an Alexa Fluor 488-conjugated goat anti-rabbit IgG (Thermo Fisher). For PicoGreen staining, cells were incubated with the Quant-iT PicoGreen dsDNA reagent (Thermo Fisher) overnight at 4°C. Fluorescence images were observed under an LSM 510 META confocal laser scanning microscope equipped with epifluorescence and an LSM digital image analyzer (Carl Zeiss, Zana, Germany). DAPI (Molecular Probes, Eugene, OR, USA) was used as a counter staining probe to mark the nuclei.

## Cellular fractionation

Cells were biochemically fractionated using a subcellular protein fractionation kit (Thermo Fisher Scientific, USA) according to the manufacturer's instructions. Briefly, cells were harvested with trypsin-EDTA, centrifuged at 500 $\times$ g for 5 min and washed with ice-cold PBS. After adding the CEB buffer to the cell pellet, the tube was incubated at 4°C for 10 min with gentle mixing. Following centrifugation at 500 $\times$ g for 5 min, the supernatant (the cytoplasmic extract) was transferred to a clean pre-chilled tube on ice. Next, the MEB buffer was added to the pellet. The tube was briefly vortexed and was incubated at 4°C for 10 min with gentle mixing. The tube was then centrifuged at 3000 $\times$ g for 5 min and the supernatant (the membrane extract) was transferred to a clean pre-chilled tube on ice. An ice-cold NEB buffer was added to the pellet, and the tube was vortexed using the highest setting for 15 s. Following incubation at 4°C for 30 min with gentle mixing, the tube was centrifuged at 5000 $\times$ g for 5 min and the supernatant (the soluble nuclear extract) was transferred to a clean pre-chilled tube on ice. Lastly, the room temperature NEB buffer containing Micrococcal Nuclease and $CaCl_2$ was added to the pellet. The tube was vortexed for 15 s and incubated at room temperature for 15 min. After incubation, the tube was centrifuged at 16,000 $\times$ g for 5 min and the supernatant (the chromatin-bound nuclear extract) was transferred to a clean pre-chilled tube on ice.

## Plasmids and mutagenesis

Flag-tagged PAPR1 WT (PARP1-Flag; #111575) was purchased from Addgene. The Flag-tagged PARP1 R138C mutant was generated by the site-directed mutagenesis Kit (Agilent, La Jolla, CA, USA) according to the manufacturer's instructions. The plasmids were subjected to DNA sequencing for verification.

## CRISPR/Cas9-mediated PARP1 knockout (KO)

In order to knock out PARP1 via the CRISPR/Cas9 system, sgRNAs of PARP1 were designed using the CRISPR design website (http://crispr.mit.edu/) and were incorporated into the lentiCRISPR_v2 plasmid. Cells were then plated in 6-well plates and were transfected with these plasmids. After 24 hr of culture and puromycin selection (1 µg/ml), single cells were sorted into 96-well plates. After a 2 week culture period, protein lysates were extracted and PARP1 KO was confirmed by immunoblot analysis. The sgRNAs were listed (See also *Supplementary file 3*).

## Cell viability measurement

MHH-ES-1 cells were plated into 96-well plates at densities of 1000 cells/well. One day later, cells were treated with various PARP inhibitors and iRucaparib-AP6 as indicated. Cell viability was measured using the CellTiter-Glo assay (Promega) according to the manufacturer's instructions. Briefly, after incubation, room temperature CellTiter-Glo reagent was added 1:1 to each well and the plates were incubated at room temperature for 2 min. Luminescence was measured with the Synergy HT Multi-Detection Microplate Reader and was normalized against control cells treated with DMSO.

## RNA interference and transfection

To produce the lentiviruses, shRNA plasmids were co-transfected into HEK293TD cells along with packaging (Δ8.9) and envelope (VSVG) expression plasmids using the Lipofectamine 2000 reagent (Invitrogen) according to the manufacturer's instructions. The next day, the media was refreshed. After two days, viral supernatants were collected and filtered using a 0.45 $\mu$m filter. Recipient cells were infected in the presence of a serum-containing medium supplemented with 8 µg/ml Polybrene. Two days after infection, cells were used for the indicated experiments. Lipofectamine 2000 reagents were also used to transiently knock-down or over-express the target genes, according to the manufacturer's instructions. Two days after infection or transfection, the cells were used for the indicated experiments. The knock-down or over-expression of target genes was validated by immunoblot assays. The following shRNA constructs and over-expression plasmids were used (See also *Supplementary file 3*). The cGAS knockdown for RNA interference was achieved using Mission shRNA-encoding lentivirus directed to human cGAS mRNA (Sigma; GenBank/EMBL/DDBJ accession no. NM_138441) as recommended by the manufacturer's protocols. Briefly, lentiviral vectors (in pLKO.1) containing cGAS shRNA sequences (shcGAS #1, TRCN0000428336; shcGAS #2, TRCN0000149811) and non-target shRNA control vector (shScramble, SHC016) were purchased from Sigma.

## Real-time quantitative polymerase chain reaction (RT-qPCR)

The mRNA extraction was performed using the RNeasy Mini Kit (QIAGEN) according to the manufacturer's instructions. Subsequently, total RNAs were converted into cDNA using the SuperScript III Reverse Transcriptase (Thermo Fisher Scientific) following the manual for first-strand cDNA synthesis. qPCR reactions were performed on a CFX384 Touch Real-Time PCR Detection System using 2X Power SYBR Green PCR Master Mix (Thermo Fisher Scientific). For each condition, technical triplicates were prepared and the quantitation cycle (Cq) was calculated. For normalization, GAPDH levels were used as an internal reference and the relative expression levels were presented. The primers used in qPCR were listed (See also *Supplementary file 3*).

## Sample preparation for mass spectrometry

MHH-ES-1 cells were treated with or without Talazoparib (0.1 or 1 µM) for 24 hr. Cells were lysed with 1% SDS lysis buffer containing 10 mM HEPES, pH 7.0, 2 mM $MgCl_2$, 20 U/mL universal nuclease. Protein concentrations were determined with the BCA assay (Thermo Fisher Scientific). Samples from two biological replicates were reduced with 3 mM dithiothreitol (DTT) for 20 min and were alkylated with 25 mM iodoacetamide (IDA) for 30 min at room temperature (RT) in dark. The detergents were removed by methanol/chloroform precipitation. The proteins were re-solubilized in 8 M urea and digested by Lys-C at a 1:100 (w/w) enzyme/protein ratio for 2 hr, followed by trypsin digestion at a 1:100 (w/w) enzyme/protein ratio overnight at RT in 2 M urea. The peptides were desalted using Oasis HLB solid-phase extraction (SPE) cartridges (Waters) (*Erickson et al., 2015*) and approximately 100 µg of peptides from each sample were re-suspended in 200 mM HEPES, pH 8.5. The

peptides were then labeled with the amine-based TMT 6-plex reagents (Thermo Fisher) for 1 hr at RT. Hydroxylamine solution was added to quench the reaction and the labeled peptide samples were combined. The TMT samples were lyophilized and reconstituted in buffer A (10 mM Ammonium formate, pH 10.0). Samples were centrifuged at 10,000 $\times$ g for 3 min using spin-X centrifuge tube filters (Thermo Fisher Scientific) prior to loading onto a ZORBAX 300Extend-C18 HPLC column (Agilent, Narrow Bore RR 2.1 mm x 100 mm, 3.5 µm particle size, 300 Å pore size). Peptides were fractionated by bRPLC (basic pH reversed phase HPLC) at a 0.2 mL/min flow rate using a gradient from 0% to 70% buffer B (1% Ammonium formate, pH 10.0% and 90% Acetonitrile). A total of seventeen fractions were collected, which were lyophilized, desalted, and analyzed by LC-MS/MS experiments.

## Quantitative proteomic analysis by LC-MS/MS

The TMT samples were analyzed on a Q-Exactive HF Mass Spectrometer (Thermo Fisher Scientific). MS/MS spectra were searched against a composite database of human protein sequences (Uniprot) and their reversed complement using the Sequest algorithm (Ver28) embedded in an in-house-developed software suite (*Huttlin et al., 2010*). MS1 and MS2 mass tolerances were set to be 50 ppm and 0.05 Da, respectively. Search parameters allowed for full tryptic peptides (two missed cleavage sites) with a static modification of 57.02146 Da on cystine (Carbamidomethyl), a variable modification of 15.994915 Da on methionine (oxidation), and a static modification of TMT labels (229.16293 Da) on peptide N-terminus and lysine. Search results were filtered to include <1% matches (both peptide and protein level filtering) to the reverse database by the linear discriminator function using parameters including Xcorr, dCN, missed cleavage, charge state (exclude 1+ peptides), mass accuracy, peptide length, and fraction of ions matched to MS/MS spectra. Peptide quantification was performed by using the CoreQuant algorithm implemented in an in-house-developed software suite (*Erickson et al., 2017*). The labeling scheme for the TMT experiments is: 126: DMSO; 127: Talazoparib (0.1 µM); 128: Talazoparib (1 µM), 129: Talazoparib (1 µM); 130: Talazoparib (0.1 µM); 131: DMSO. For TMT quantification, a 0.03 Th window was scanned around the theoretical m/z of each reporter ion (126: 126.127726; 127: 127.124761; 128: 128.134436; 129: 129.131471; 130: 130.141145; 131: 131.138180) to detect the presence of these ions. The maximum intensity of each ion was extracted, and the signal-to-noise (SN) value of each protein is calculated by summing the reporter ion counts across all identified peptides. Because the same amount of peptides was used for each TMT channel, the total reporter ion intensity of each channel was summed across all quantified proteins, and was then normalized and reported. Data were exported to Excel for further analysis.

## Statistics

All statistical analyses including unpaired Student's t-tests, one- and two-way ANOVA tests were performed using the GraphPad Prism software (v8.2.0). The type of statistical analyses, parameters, and number of replicates are indicated for each experiment in the figure legends. Data were calculated as mean ± SEM or SD. The following indications of significance were used throughout the manuscript: $^*p < 0.05$, $^{**}p < 0.01$, $^{***}p < 0.001$, $^{****}p < 0.0001$, n.s, not significant.

## Acknowledgements

We thank Dr. Shuai Wang for the initial help with the preparation of the mass spectrometry samples. This work was supported, in part, by grants from the NIH (R01GM122932 and R35GM134883 to YY) and Welch foundation (I-1800 to YY).

## Additional information

#### Competing interests

Yonghao Yu: A patent application on the PARP degraders was previously filed (PCT/US2020/016129). The other authors declare that no competing interests exist.

### Funding

| Funder | Grant reference number | Author |
| --- | --- | --- |
| National Institute of General Medical Sciences | R01GM122932 | Yonghao Yu |
| National Institute of General Medical Sciences | R35GM134883 | Yonghao Yu |
| Welch Foundation | I-1800 | Yonghao Yu |

The funders had no role in study design, data collection and interpretation, or the decision to submit the work for publication.

### Author contributions

Chiho Kim, Conceptualization, Data curation, Formal analysis, Methodology, Writing - original draft, Writing - review and editing; Xu-Dong Wang, Resources, Data curation, Formal analysis, Methodology; Yonghao Yu, Conceptualization, Formal analysis, Supervision, Funding acquisition, Investigation

### Author ORCIDs

Chiho Kim  https://orcid.org/0000-0001-6846-5515
Xu-Dong Wang  http://orcid.org/0000-0002-8265-1485
Yonghao Yu  https://orcid.org/0000-0001-8414-4666

### Decision letter and Author response

Decision letter https://doi.org/10.7554/eLife.60637.sa1
Author response https://doi.org/10.7554/eLife.60637.sa2

## Additional files

### Supplementary files

- Supplementary file 1. Raw and analyzed TMT-MS data in MHH-ES-1 following Talazoparib treatment (1 µM for 24 hr).
- Supplementary file 2. GO analysis of up-regulated proteins from *Supplementary file 1*.
- Supplementary file 3. Oligo sequence in this study.
- Transparent reporting form

### Data availability

The raw and analyzed TMT-MS data in MHH-ES-1 cells following Talazoparib treatment is provided in Supplementary file 1. Its GO analysis data using up-regulated proteins is provided in Supplementary file 2.

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
