## [Decision Letter]

**Acceptance summary:**

We believe your study provides an important understanding to the mechanism of action of PARP1 inhibitors. Such understandings may in turn trigger the develop of new PARP1 inhibitors for therapeutic purposes.

**Decision letter after peer review:**

Thank you for submitting your article "PARP1 inhibitors trigger innate immunity via PARP1 trapping-induced DNA damage response" for consideration by *eLife*. Your article has been reviewed by three peer reviewers, one of whom is a member of our Board of Reviewing Editors, and the evaluation has been overseen by Philip Cole as the Senior Editor. The reviewers have opted to remain anonymous.

The reviewers have discussed the reviews with one another and the Reviewing Editor has drafted this decision to help you prepare a revised submission.

Summary:

The manuscript provides compelling evidence to show that trapping PARP1 on chromatin by PARP inhibitors is important for inducing cGAS_STING. This includes the use of PARP1 knockout cells, a PARP1 mutant with an altered trapping ability, and PARP1 inhibitors with different trapping activity. The paper is well written and was enjoyable to read. The results will be important for the field and thus should be published after minor revisions as described below.

1) The authors should test the different PARP1 inhibitors, including the PROTAC version, in relevant cancer cell lines to find out whether the trapping ability is important for the anticancer activity of PARP1 inhibitors.

2) Please provide proper loading controls to the blots where these are not shown.

3) Please check the existing proteomic data for the cytokines that have been examined at mRNA levels and provide this information in the revised manuscript.

4) Please add to the Discussion possible mechanisms by which PARP1 trapping increases cytosolic DNA and cite and briefly connect their results to the following paper: Zandarashvili et al., 2020.

Revisions expected in follow-up work:

Testing inhibitors of different trapping ability in mouse models of cancer would be interesting to follow up in future studies.

---

## [Author Response]

Revisions for this paper:1) The authors should test the different PARP1 inhibitors, including the PROTAC version, in relevant cancer cell lines to find out whether the trapping ability is important for the anticancer activity of PARP1 inhibitors.

We thank the reviewer for this important point. The cytotoxic effects of PARP1 inhibitors (PARPi) can be ascribed to two distinct, yet interconnected mechanisms, i.e., PARP1 inhibition and PARP1 trapping. We surveyed a series of clinically relevant PARPi, including Talazoparib, Niraparib, Rucaparib, Olaparib, and Veliparib. Consistent with previous studies, these compounds all potently blocked the enzymatic activity of PARP1 (Figure 3—figure supplement 1A). However, these compounds were able to induce different levels of PARP1 trapping (PARP1 trapping levels: Talazoparib > Niraparib > Rucaparib ≈ Olaparib > Veliparib) (Figure 3A). Interestingly, DDR as measured by γH2AX was correlative with respect to the level of PARP1 trapping elicited by these PARPi (DNA damage levels: Talazoparib > Niraparib > Rucaparib ≈ Olaparib > Veliparib) (Figure 3B). We have now measured the cytotoxicity of these PARPi in MHH-ES-1, which is a PARPi-sensitive Ewing’s Sarcoma cell line. We found that the cytotoxicity of these compounds also positively correlated with their PARP1 trapping activities (data provided in Figure 3—figure supplement 1D).

To further assess the relative contribution of PARP1 inhibition vs. PARP1 trapping in mediating the cytotoxicity of PARPi, we used iRucaparib-AP6, which is a specific PARP1 PROTAC compound. By degrading PARP1, iRuccapaib-AP6 blocks the enzymatic activity of PARP1, without eliciting PARP1 trapping. We showed that iRucaparib-AP6 led to robust downregulation of PARP1. In contrast, the parent compound (Rucaparib) only induced the cleavage, but not the degradation of PARP1, presumably because of its toxicity in these cells (Figure 4A). Using a different cell line system (i.e., MHH-ES-1), we also showed that Rucaparib, but not iRucaparib-AP6, strongly promoted cell death (data provided in Figure 4—figure supplement 1).

Taken together, these data indicate that PARP1 trapping is the major driver of the cytotoxicity of PARPi.

2) Please provide proper loading controls to the blots where these are not shown.

We have now added the proper loading controls to the blots (Figure 1D, Figure 1—figure supplement 1G, Figure 2C, Figure 3C, and Figure 4D).

3) Please check the existing proteomic data for the cytokines that have been examined at mRNA levels and provide this information in the revised manuscript.

Using q-RT-PCR assays, we found that Talazoparib treatment in MHH-ES-1 cells greatly induced the expression of innate immune-related genes, including type I interferons (IFN; Inf-α and Inf-β), pro-inflammatory cytokines (Ccl5 and Cxcl10), and interferon-stimulated genes (ISGs; Isg15, Mx1, Mx2, and Ifit3).

We have now gone back to our quantitative proteomic data, and extracted these information. Specifically, we found Interleukin 1*α* (IL1A) as one of the most up-regulated cytokines in Talazoparib-treated MHH-ES-1 cells (Supplementary file 1). In addition, we also identified several up-regulated proteins that are related to innate immune response, including ISG15, IFIT3, MX1 and MX2. Together, these results support that PARPi correlatively up-regulates both genes and proteins which are related to innate immune response.

Taken together, our data show that PARPi treatment in MHH-ES-1 cells upregulates the expression of innate immune related genes at both the mRNA and protein levels. We have now added these discussions (Results).

4) Please add to the Discussion possible mechanisms by which PARP1 trapping increases cytosolic DNA and cite and briefly connect their results to the following paper: Zandarashvili et al., 2020.

We thank the reviewer for raising this important point. In this paper, we have shown that PARP1 trapping is major driver for PARPi-mediated DDR, cytotoxicity and innate immune signaling. We utilized a panel of 5 clinically relevant PARPi (i.e., Talazoaprib, Niraparib, Rucaparib, Olaparib, and Veliparib) Lord and Ashworth, 2017(). While these compounds all potently blocked the formation of PAR, they displayed a dramatically different capability in inducing PARP1 trapping. Indeed, a recent study showed that the different structural elements within these PARPi drive, in an allosteric manner, the release or retention of these compounds at a DNA break Zandarashvili et al., 2020(). Intriguingly, these compounds then induce DDR and activation of cGAS-STING signaling that is closely correlated with the degree of PARP1 trapping (Figure 3A-D, Figure 3—figure supplement 1A and D).

Although the detail mechanisms remain unknown, recent studies have pointed to cytosolic dsDNA as a source of stimulus for the activation of cGAS-STING pathway Sun et al., 2013Wu et al., 2014Civril et al., 2013(, , ). In the context PAPRi, cytosolic dsDNA could result from PARPi-induced PARP1 trapping, and subsequently, stalled or collapsed replication forks. The detailed molecular underpinnings of this critical pathway will be addressed in future studies. We have added these discussions in the manuscript (Discussion).